# Novel Magnetic Nano Silica Synthesis Using Barley Husk Waste for Removing Petroleum from Polluted Water for Environmental Sustainability

**Evidence Akhayere** [1,2], **Ashok Vaseashta** [3,4] **and Doga Kavaz** [2,5,*]

1   Department of Environmental Science, Cyprus International University, Northern Cyprus via Mersin 10, 98258 Nicosia, Turkey; eakhayere@ciu.edu.tr
2   Department of Bioengineering, Cyprus International University, Northern Cyprus via Mersin 10, 98258 Nicosia, Turkey
3   International Clean Water Institute, Manassas, VA 20112, USA; prof.vaseashta@ieee.org
4   Biomedical Engineering and Nano technologies Institute, Riga Technical University, 1658 Riga, Latvia
5   Environmental Research Centre, Cyprus International University, Northern Cyprus via Mersin 10, 98258 Nicosia, Turkey
*   Correspondence: dkavaz@ciu.edu.tr

**Abstract:** Water contamination by petroleum and its byproducts presents a major challenge worldwide. It is critical that sustainable treatment methods be employed for the removal of such contaminants from polluted water. For this investigation, magnetic nano silica (M-NS) was synthesized using agricultural waste from barley husk using a two-step process that is environmentally friendly and uses green chemistry synthesis. The barley husk waste was used as a precursor for the synthesis of nano-silica following a low energy and sustainable method of acid reflux and heat treatment. Nano-silica was then used for the synthesis of M-NS, with the addition of a magnetic solution of $Fe_3O_4$ nanoparticles. The magnetic nano-silica particles were characterized using scanning electron microscopy (SEM), Fourier transform infrared (FTIR), Zeta potential analysis (ZETA) and X-Ray Diffraction (XRD). Magnetic nano-silica particles were observed to have an average diameter of 162 nm and appeared to be hydrophobic, with a large surface area of ~120 $m^2$/gm. Due to these characteristics, magnetic nano-silica was used as an adsorbent for the removal of petrol contaminants from water. The experimental procedure showed that only 0.6 gm. of M-NS was used on 40 mg/L concentration of petroleum and the experiments recorded a high uptake efficiency of 85%. The sorption was shown to be an effective process since a high amount of petroleum was removed. The study further demonstrates that as the amount of sorbent is increased, the sorption capacity also increases until an equilibrium is reached. The results of this study establish that synthesis of M-NS, using environmentally sustainable processes, has the required characteristics to serve as sorbent for petroleum and its byproducts from contaminated water, thus enhancing environmental sustainability.

**Keywords:** nano silica; sorbent; petroleum; environment; sustainability; barley husk waste

---

## 1. Introduction

In our environment, waste materials are generated from different sectors of society on a daily basis and one of the sectors that generate a high amount of waste is the agriculture sector. There are several categories of waste generated from the agricultural sector and notable examples include animal manure, banana peels, rice husk, food fiber, etc. [1–3], to name just a few. Barley is a plant that is largely cultivated in Cyprus and across the Mediterranean region and generates the largest amount of agricultural wastes in these areas. Barley husk (BH) is an example of an agricultural waste product [4,5]

obtained during the harvesting season due to usage of these plants [6]. In Mediterranean areas like North Cyprus, where agriculture is a very large part of everyday life, there is a very high rate of the generation of barley waste, linked to a high rate of cultivation. This is also true for other agriculturally intensive countries. The farmers, being unaware of potential uses of such wastes, leave barley waste unattended, thereby creating a tremendous environmental burden [7]. Barley plant belongs to the grass family and it is the fourth-largest grain crop grown around the world. As a grass plant, there is close contact between the plant and the soil, which causes a high silicon presence in the plant [8]. In research reported by Akhayere et al. [9] and Azizi et al. [10], X-ray fluorescence (XRF) results showed that silicon based compounds were abundantly present in barley, which makes it a good raw material for the production of silicon-related substances. To produce silica and other silicon related materials from conventional sources, one would require high amounts of energy due to the temperature requirements, ranging from ~1300 °C upwards. In addition to high energy demand, the process produces large amounts of $CO_2$ gas [10–13]. It was shown by Chen et al., 2016 [14], that nano-silica is a very good sorbent for the removal of oily contaminants from water. In general, nanoparticles have proven to be good adsorbents due to their large surface area [15–17]. Nano silica is a very interesting nanoparticle because it is a chemically inert with no adverse health implications, which makes it a good material for water contamination treatment [18].

Crude oil and petroleum products remain highly utilized sources of energy throughout the world and will continue to do so for many years to come. The exploration of crude oil reserves, storage and transportation of oil and petroleum products are accompanied by a high risk of leak and spillage which results in environmental damage [19,20] and the associated high cost of cleanup. Water contamination by petroleum products has been a problem of major concern to environmental scientists and engineers after major oil spills, such as Exxon Valdez, Gulf War and leaks at extraction sites, such as Deepwater Horizon and Taylor Energy. It is imperative that scientists and environmentalists need to develop cost-effective materials for the treatment of water contaminated oil and petroleum products [21–23] to enhance environmental sustainability. Different responses and remedy methods have been introduced, and each of these methods have played their role in the reduction in the amount of damage that crude oil contamination can cause [24,25]. In addition, the release of long-chain hydrocarbons in water streams have also been identified as an emerging contaminant of concern.

The use of consumable plants-based waste for synthesis and production of materials is known to be highly sustainable, as it poses little to no negative impact on natural resources, the environment, organisms and people, although some have argued that the impact of agricultural waste on human health and the environment cannot be compared to that of conventional waste, but sustainable ideologies are geared towards the lowest or possibly zero-waste generation [26–28]. Subsequently, the recycling, upcycling or reuse of agricultural waste materials can help save the environment from depletion and degradation, owing to the fact that materials produced from agricultural waste cause very little and sometimes no amount of stress on the environment [29–31].

In this study, we show a sustainable transformation of barley husk waste into magnetic nano silica and its application for the removal of petroleum from water, which is equally applicable to other long-chain hydrocarbon products, resulting in an environmentally sustainable outcome. Several studies have explored the use of chemically produced nano silica as an adsorbent for the removal of oily contaminants in water, but in terms of sustainability of the production of nano silica, these processes do not meet the green chemical synthesis paradigm. Nano silica synthesized in this study was achieved through the green synthesis paradigm, including the raw material for its synthesis, which was also geared towards achieving a sustainable environment. The magnetization of the nano silica allowed easy collection of the adsorbent after adsorption, by the simple use of a magnet.

As described below, the objective of the investigation was to synthesize magnetic nano silica using a two-step environmentally-friendly process. The magnetic nano silica particles were characterized for their size, surface area, morphology, magnetization, adsorption and reusability. The magnetic nano silica was then used for the removal of petroleum-based contaminants from water. The application of

silica nanoparticles (SNPs) cannot be overemphasized, since SNPs are now being increasingly used in various other industrial applications, thus providing an additional basis for this research as an alternative route to meet the growing need of silica by industries.

## 2. Materials and Methods

Nano silica used in this study was synthesized using barley husk waste. The gasoline used in this experiment was purchased from a local petrol station. For characterization of synthesized nano-silica, SEM, XRD, FTIR and ZETA potential characterization equipment were available for use in the Cyprus International University Environmental Laboratory. Other materials such as a furnace, miller equipment, reflux bottles, beakers, and hydrochloric acid are all standard scientific laboratory supplies and instruments and were obtained from the same laboratory.

### 2.1. Synthesis of Nanoparticles

The barley husk was cut into smaller pieces and dried for seven days. The pieces were washed thoroughly using distilled water to remove any adhering sand, soil and other visible particles from the straws. The resulting sample was dried in the oven for 24 h. at 100 °C. The dried barley husk was then crushed to powder form using a miller. A quantity of 50 gm. of barley husk powder was then refluxed in 250 mL of 2 M HCl for 6 h. After the acid refluxing, the sample was filtered and washed in distilled water, followed by heating at 700 °C for 5 h. The resulting sample was dissolved in 10% $HNO_3$, for 3 h and then rinsed with distilled water, filtered and allowed to cool, during which nano silica precipitated. Heating helped to remove the organic content while acid reflux helped to remove metallic impurities from the barley. The synthesis procedure for producing nano-silica, as shown in Figure 1, is also described elsewhere by Akhayere et.al. [9]. The resulting nano silica powder was used for the preparation of nano silica solution.

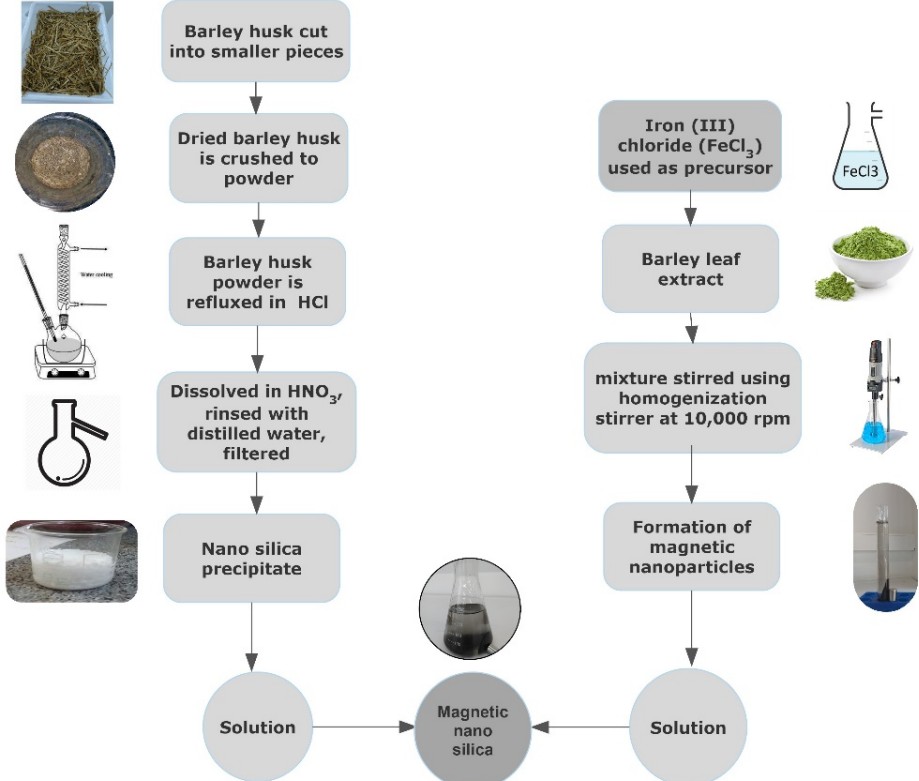

**Figure 1.** Schematic diagram of the experimental procedure.

To synthesize Magnetic nano silica (M-NS), it was important to prepare the magnetic solution, as this step is followed by mixing with nano silica solution. Iron (III) chloride (FeCl$_3$) was used as the precursor for the synthesis of the magnetic nanoparticles. A quantity of 20 mL of the barley leaf extract/powder was added to 200 mL of 1 mM FeCl$_3$ solution at room temperature. The resultant mixture was stirred using a homogenization stirrer at 10,000 rpm for 60 min [32,33]. During this process, the color of the mixture changed to a very dark brown as seen in Figure 2b, indicating the formation of magnetic nanoparticles. The mixture was later centrifuged three times for 10 min each and washed with alcohol twice. The color of the mixture became progressively darker, as can be seen in Figure 2b, during the process. By bringing a magnet in close proximity, the magnetic behavior of the particles was clearly evident, as is shown in Figure 2c.

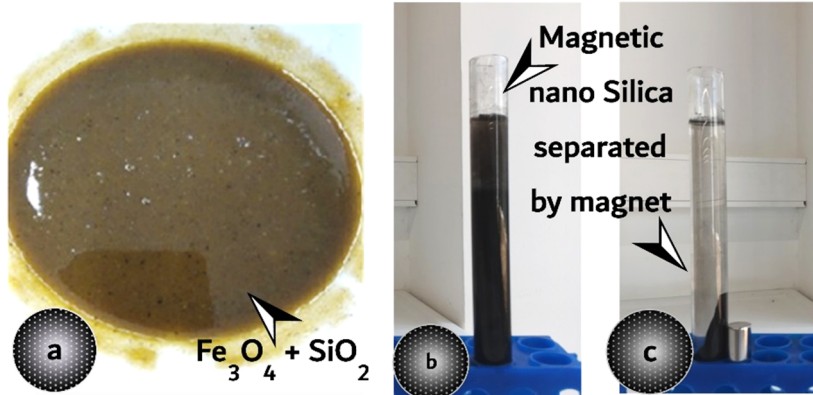

**Figure 2.** Synthesis route of M-NS. (**a**): Fe$_3$O$_4$ + SiO$_2$ solution, (**b**): homogenized mixture, and (**c**): silica nano particles centrifuged and separated by a magnet.

Magnetic nano silica (M-NS) particles were synthesized by slowly dropping the magnetic solution into the nano-silica solution with continuous mixing for 15 min, after which the solution was centrifuged (Figure 2a) and dried in the oven at 80 °C for 3 h.

## 2.2. Characterization

To be able to understand the behaviors of the synthesized magnetic nanoparticle, XRD patterns were measured with an X-ray diffractometer (Bruker D8 Advance model). Subsequently, the FTIR spectra were carried out using (IR Prestige, 21 Shimadzu, Japan) to investigate the significant FTIR peaks and variations, and significant peaks were observed and recorded over the range of 400–4000 cm$^{-1}$. This analysis was important for the determination of the functional groups that are present in synthesized magnetic nano silica. To determine the stability and particle sizes of the synthesized nanoparticles, it was necessary to conduct Dynamic Light Scattering (DLS) analysis to enable the determination of surface charge of the nanoparticles as well as the particle diameter. To understand the magnetic properties of M-NS, magnetization studies were carried out using a superconducting quantum interference device (SQUID) magnetometer. FTIR, XRD and SEM analysis were also carried out for both nano silica and iron nano particles to compare any observed changes.

### Magnetic Properties

To understand the magnetic properties M-NS of the particles, sample investigations were carried out using a SQUID magnetometer (Quantum Design MPMS XL). A quantity of 12 mg of the synthesized M-NS particles was weighed accurately and placed inside a gelatin capsule, which was immobilized by GE Varnish glue (Cryospares, Abingdon). The magnetization of the nanoparticles was measured at 300 K as a function of a magnetic field.

*2.3. Adsorption Study of Oil Removal by Magnetic Nano-Silica*

2.3.1. Adsorption Experiment

The sorption experiments were performed in batches, as shown in Figure 3a,b to ascertain the effectiveness of the synthesized magnetic nano-silica for removal of the petroleum from contaminated water by adding 0.6 g of magnetic nano-silica (M-NS) to a beaker containing 500 mL of petroleum contaminated water. The concentration of the petroleum contaminant was 40 mg/L. The effect of contact time between the nanoparticles and the contaminant was first observed; experimental contact was allowed for 2, 4, 6, 8, 10, 12, 14, 16, 18 20, and 22 min, respectively. The maximum sorption was observed after 10 min, which was the time at which equilibrium was reached. The optimization of pH was carried out by adjusting the pH of the system at different time values from 2 to 10, after which the samples were analyzed to obtain the most suitable pH value for the experiment. The amount of adsorbent used for sorption is a very important parameter to obtain the required amount of sorbent. For effective sorption of petrol, different amounts were used on the contaminated water at a neutral pH [34] with the adequate contact time, and the samples were then taken for analysis in the Oil Content Analyzer, Horiba OCMA 310, which first extracts the oil using a solvent (Polychlorotrifluoroethylene) and then analyzes it by infrared spectrophotometry. The result is given mg/l oil on a digital panel meter. The initial petrol concentration and the concentration after each analysis is used to calculate the percentage of uptake efficiency using the formula, as provided below. The uptake efficiency % is $= \left( \dfrac{C_0 - C_1}{C_0} \right) \times 100$, where $C_0$ is the initial concentration and $C_1$ is the final concentration.

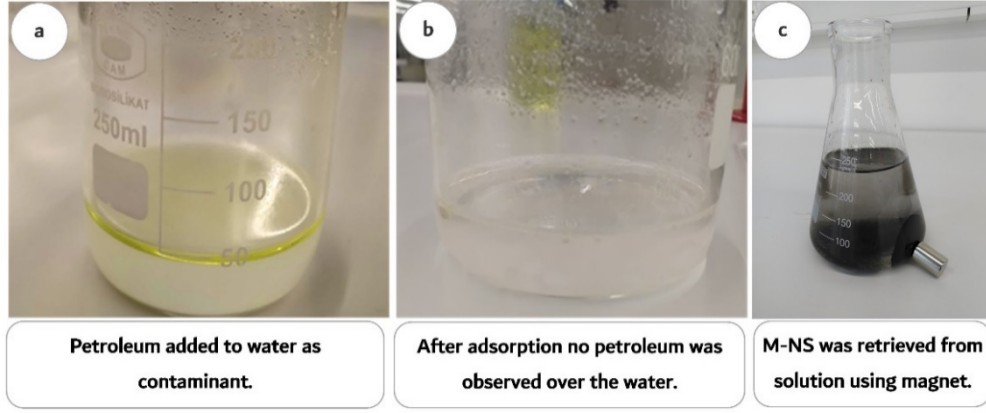

**Figure 3.** (**a**) Contaminated water by petrol, before adsorption. (**b**) Water sample after adsorption. (**c**) M-NS in acetic acid for desorption.

2.3.2. Optimization of Contact Time

The effect of contact time for the sorption experiment was studied by using 0.6 gm of synthesized magnetic nano-silica (M-NS) in a beaker containing 500 mL water contaminated by 40 mg/L petroleum. The sample was adjusted at optimized pH (pH = 7) and withdrawn at different time intervals (2, 4, 6, 8, 10, 12, 14, 16, 18, 20 and 22 min).

2.3.3. Optimization of pH

To understand the behavior of the sorption concerning changes in pH, 0.6 g of synthesized M-NS was used as a sorbent for a beaker containing 500 mL water contaminated by 40 mg/L petroleum. The reaction pH was adjusted at each withdrawal interval between pH 2 and 10.

*2.4. Desorption*

The reusability of the M-NS was assessed by conducting desorption studies. Acetic acid was used as the optimum solvent for the desorption experiments. Subsequently, the experiment was carried out by mixing the exhausted M-NS in 5 mL acetic acid solution, and afterwards the mixture was shaken in a water bath for 85 min at 35 °C; this procedure was repeated after each use. In Figure 3c, it can be seen that, as a result of the magnetic properties of nanoparticles, the M-NS were easily washed in the acetic acid solution and retrieved using a magnet. The reusability of the magnetic nanoparticles after adsorption of contaminants is another indicator of the sustainability of the materials.

## 3. Results and Discussion

*3.1. Characterization Results of Magnetic Silica Nanoparticles*

The X-ray pattern of magnetic nano-silica (M-NS) particles is shown in Figure 4a. The diffraction peaks of synthesized $Fe_3O_4$-NPs shown in Figure 4a-C were detected at $2\theta$ = 30.6°, 35.9°, 43.5°, 54.0°, 57.4°, 63.00°, and 74.5°; these were allocated to the crystal planes of (200), (311), (400), (422), (511), (440), and (533), respectively. The diffraction peaks as analyzed, corresponded with the standard magnetite XRD patterns according to the Joint Committee on Powder Diffraction Standards (JCPDS) file #: 00-003-0863, for the cubic structure crystallographic system. We can calculate the crystallite size of the synthesized $Fe_3O_4$-NPs with the help of the Debye–Scherrer equation, which reveals a relationship between X-ray diffraction peak broadening and crystallite size [35]. The estimated crystallite size of synthesized $Fe_3O_4$-NPs was 10.12 nm; this is calculated from the full-width at half maximum of the crystal planes of (311) [36]. According to the XRD pattern, the synthesized $Fe_3O_4$-NPs were seen to be pure crystalline lean with little or no notable impurity peaks.

Subsequently, the XRD peak of nano silica was studied at $2\theta$ = 20–30°. The result, as shown in Figure 4a-B, reveals that nano silica particles were observed to be amorphous; however, the result also revealed that there were very few impurities as a result of other chemical elements present in the barley husk waste, which was used as the raw material for synthesis of nano silica. The highest peak of the nano-silica particle was observed at $2\theta$ = 24.8°. According to the XRD results, it is valid to say that the method of synthesis of nano-silica from barley resulted in particles with significantly low impurities [9,16]. Lastly, it can be seen from the XRD result that there were no substantial changes observed in case of the peaks for magnetic nano silica, as shown in Figure 4a-A, although a new peak was identified at $2\theta$ = 20–30°, which indicates the presence of amorphous silica. Similar results were observed in XRD patterns obtained by Munasir and Terraningtyas, 2019 [37].

Figure 4b shows the FT-IR spectroscopy of the nanoparticles. FTIR was performed to understand the functional groups present in the nanoparticles. Barley leaf extract played the role of stabilizing and capping agent during the synthesis of the magnetic nanoparticles. The two peaks observed at 560 and 423 $cm^{-1}$ in the spectra of synthesized $Fe_3O_4$-NPs, as observed in Figure 4b-C, can be attributed to the stretching vibration mode of Fe–O. Subsequently, the metal oxygen band shown at 560 $cm^{-1}$ is related to the stretching vibrations of metals at the tetrahedral site, while the octahedral–metal stretching of Fe–O showed a metal–oxygen band found at 423 $cm^{-1}$ [38]. The FTIR of nano silica synthesized from barley husk waste is seen in Figure 4b-B. In a given range of 600–4000 $cm^{-1}$, the most dominant peak can be seen at 1080 $cm^{-1}$, and this peak can be attributed to the asymmetrical stretching vibrations that correspond with silicon bonding (Si-O-Si). Moreover, the peak at 2200 $cm^{-1}$ was associated with the C=H stretching of an aromatic methyl group. Furthermore, the peak at 800 $cm^{-1}$ indicated siloxane bonds (Si-O-Si) forming silica from silicon, which indicates that the results obtained in this study are congruent with those of previous reported research [39]. FTIR for magnetic nano silica synthesized from barley husk investigation is shown in Figure 4b-A.

It is observed that there were significant peaks and bends as a result of the vibrations of the functional elements. The peak at 980 $cm^{-1}$ indicates the siloxane bond (Si-O-Si) forming silicon from silica, whereas the absorption bands at 3440 $cm^{-1}$ correspond to O–H stretching mode [9,30].

The spectra of magnetic nano-silica showed a large peak of absorption bands near 1200 cm$^{-1}$ and these bands were assigned to Fe-O-Si stretching mode. The SiO$_2$ spectrum shows a stretching vibration at 980 cm$^{-1}$ which is associated with Si–O–Si. The nano-silica spectrum shows a visible absorption around 1200 cm$^{-1}$, which can be related with Si–O stretching mode. This absorption can also be seen in the Fe$_3$O$_4$/SiO$_2$ spectrum and with a longer intensity [40,41]. The presence of SiO$_2$ formation is visible in the spectrum of Fe$_3$O$_4$/SiO$_2$ and can be explained by the Si–O–Si stretching vibration at 980 cm$^{-1}$ together with Fe–O–Si stretching vibration at 1250 cm$^{-1}$. The available results give a strong suggestion that silicon nanoparticles synthesized from barely husk waste were successfully magnetized by Fe$_3$O$_4$ nanoparticles [42].

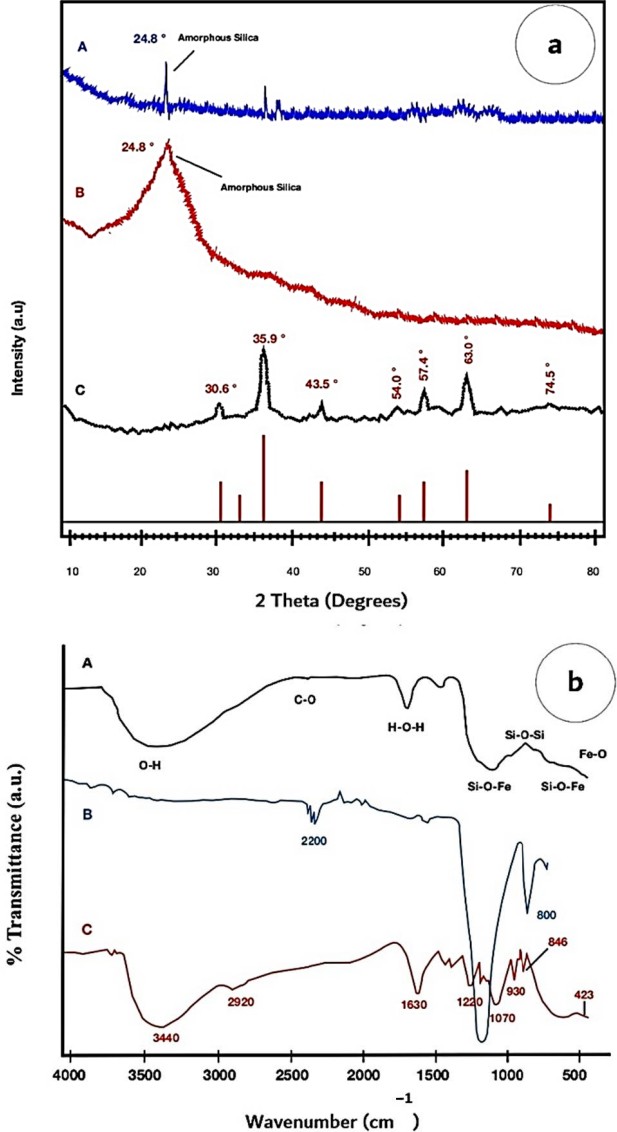

**Figure 4. (a)**: X-ray diffraction patterns of (A) Magnetic nano silica particles (M-NS), (B) NS-B particles and (C) iron oxide nanoparticles. **(b)**: FTIR spectra of (A) Magnetic nano silica particles (M-NS), (B) NS-B particles and (C) iron oxide nanoparticles.

When considering the characteristics of nanoparticles, the size distribution of the nano particle is a very important information to consider. The size of nanoparticles before and after the synthesis of magnetic nano silica were calculated. The result, as shown in Figure 5, gives the average size of the iron oxide nanoparticles as 14.5 nm, silica nanoparticles as 131.5 nm and magnetic nano silica (M-NS) particles as 162 nm. Furthermore, other researchers, such as Yew et al., used the green

method of synthesizing $Fe_3O_4$-NPs using seaweed *Kappaphycus alvarezii* and the mean particle size was 14.7 nm with the standard deviation of 1.8 nm [43]. According to Stjerndahl et al., superparamagnetic $Fe_3O_4/SiO_2$ nanocomposites were obtained in a particle size of 56 nm, but a microemulsion method [44] was used.

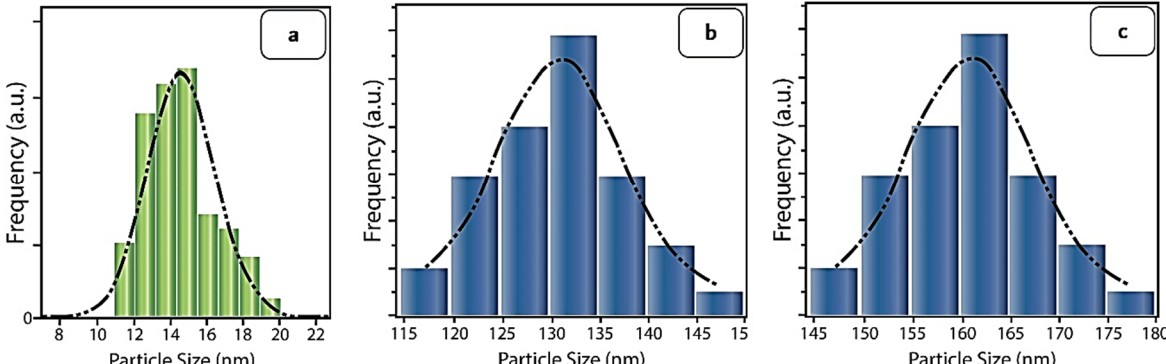

**Figure 5.** Particle size distributions of: (**a**) iron oxide nanoparticles, (**b**) NS-B particles and (**c**) Magnetic nano silica particles (M-NS).

The zeta potential stability average was also found to be 28.2 mV. Over time, nano particles have proven to be good sorbents, primarily due to their large surface area. The zeta potential of the nanoparticles shows how adequately nanoparticles could form stable structures with respect to the colloidal phase. At low zeta potentials (near zero), particles are no longer repulsed firmly, and colloids will aggregate because of surface forces of attraction. Then again, stable dispersion is obtained at high zeta potentials (above ~30 mV). This is of specific significance in water treatment applications as well as other applications where stable colloidal frameworks are required [45].

Figure 6 shows the schematic of the magnetic properties of the synthesized magnetic nano silica, showing that the highest magnetization saturation (MS) value was measured for the magnetic nano silica particles. The MS decreased when the amount of $SiO_2$ in the nano particle mixture was increased, but, with increase in $Fe_3O_4$, MS increased. Saturation occurs as a result of ferromagnetic materials, such as $Fe_3O_4$, for this investigation. As the $SiO_2$ content increased, there was an increase in ratio between magnetic remanence and magnetization saturation (MR/MS) as well as increase in coercivity ($H_C$) [46,47]. The modification of iron nano particles with silica layers would have a drastic impact on the surface properties of the nanoparticles.

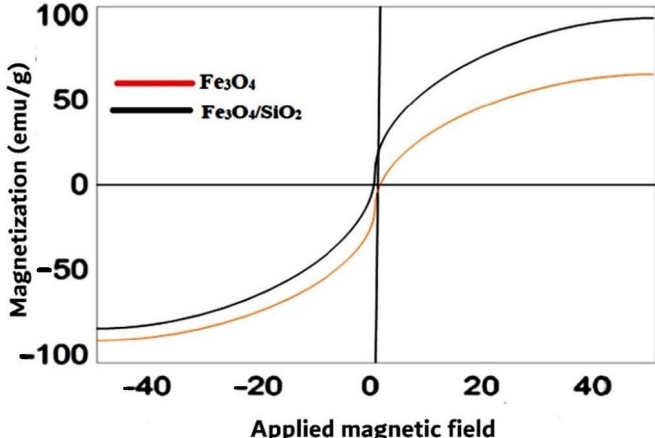

**Figure 6.** Magnetization curves of $Fe_3O_4$ and Magnetic nano silica (M-NS) particles at room temperature.

Figure 7 shows the results of SEM for nano silica, iron nano particles and magnetic nano silica (M-NS). SEM results aid in the morphological expression of nano particles. In Figure 7a, nano silica

shows some distinct spherical shapes with very little agglomeration, whereas Figure 7b shows that the morphological expression of $Fe_3O_4$ nano particles produced some agglomerated spherical structures. Figure 7c shows large agglomeration of spherical particles, suggesting that there is high presence of van der Waals forces on the surface of the particles. Although the particles in Figure 7c showed no presence of pores, the aggregated arrangements of the particles would allow contaminants to be adsorbed at different layers. This agglomeration of the magnetic nano silica particles can be attributed to the high agglomeration observed in iron oxide nano particles.

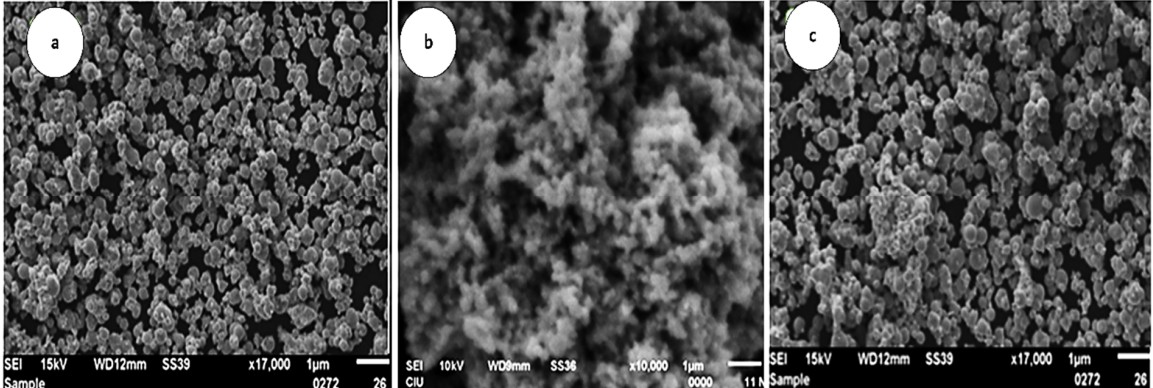

**Figure 7.** SEM results (**a**) synthesized nano silica particles, X17,000, 10kV, (**b**) synthesized iron nano particles, X10,000, 10 kV, and (**c**) magnetic nano silica M-NS. X17,000, 15 kV.

*3.2. Sorption Results*

3.2.1. Effect of Contact Time

The studied effect of contact time, as seen in Figure 8a, shows that the uptake of petrol contaminants began at 2 min at a rate of 20%, while higher sorption efficiency began at 5 min. At 8 min, the contaminant uptake increased to 65% and 80% at 10 min. The highest uptake of 85% was achieved at 12 min and the rate was constantly maintained until the experiment was concluded at 22 min. This shows that the equilibrium time for sorption of petrol contaminants using magnetic nano silica is 12 min. In experiments conducted by [48], similar sorption equilibrium times were attained. The progression of the sorption experiment in relation to time shows that sorption increased as time increased until the equilibrium time was reached.

3.2.2. Effect of pH

The effect of pH value of the solution on the effectiveness of removal of petrol was investigated and reported according to [49], since pH can affect and alter the uptake capacity of contaminants by adsorbents. In the result shown in Figure 8b, low percentages of petroleum contaminants were adsorbed by Nano silica at low pH between 4 and 6. An increment in the uptake of contaminants began to occur at pH 6 and the uptake reached its highest point at pH 7. However, when the pH was further increased, the uptake efficiency again began to drop and a large declination of uptake was observed between pH 8 and 10; the optimum pH for the adsorption of petroleum contaminants by M-NS is pH 7 in this experiment, and this is because the pH affects the charges of the sorbent and the sorbate, thereby causing an alteration between the interactions of sorbent and sorbate [50].

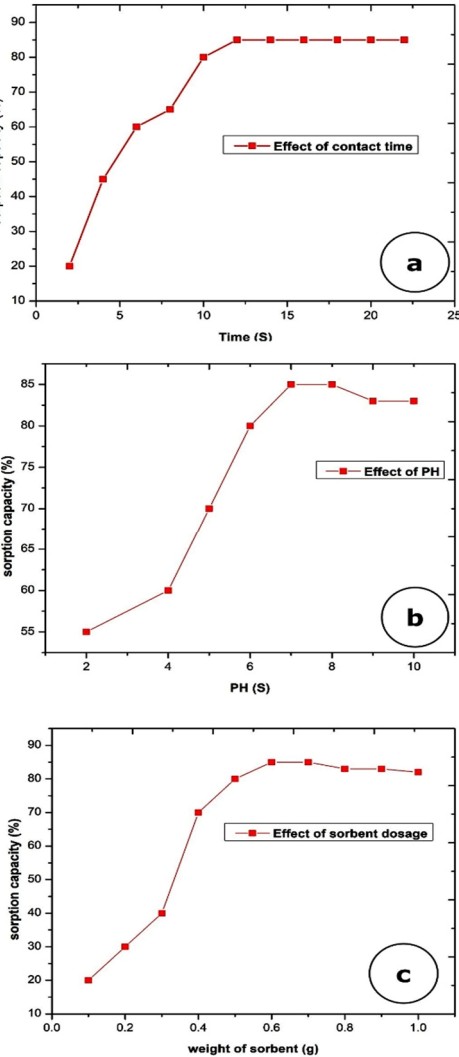

**Figure 8.** Effect of adsorption parameters on petroleum removal over M-NS particles: (**a**) contact time, (**b**) pH and (**c**) sorbent dosage.

### 3.2.3. Effect of Sorbent Weight

The availability of sites for bonding during sorption reactions is a very important factor to consider, since it is related to the amount of sorbent used in the sorption. The weight of sorbent with respect to the amount of petrol adsorbed was recorded and is as shown in Figure 8c. It is observed that the highest uptake result was obtained at 0.6 gm. sorbent weight, as shown in recent studies [51,52], which suggests that sorption increases with respect to the weight of the sorbent. The results of this study show that the sorption increases with respect to the amount of sorbent available. The sorption rate also reduced to 83% at a sorbent dosage of 0.8 gm., as against the 85% experienced earlier.

### 3.3. Adsorption Kinetics

The second-order adsorption kinetics can be described as:

$$\left(\frac{dq}{dt}\right) = k_1(q_e - q),\tag{1}$$

In this case, $q_e$ is described as the equilibrium amount of adsorbed contaminant gm. per unit of mass in grams of the adsorbent; subsequently, q is given as the amount of contaminant adsorbed at t,

which refers to time, and $k_1$ is the rate constant (1/min). Integration of the equation above with the limit q = 0 at time t = 0 gives:

$$\ln\left(\frac{q_e - q}{q_e}\right) = -k_1 t,\tag{2}$$

This equation can also be written as:

$$\ln(q_e - q) = -k_1 t + \ln(q_e),\tag{3}$$

From Equation (3), it can be predicted that a plot of $\ln(q_e - q_t)$ would result in a straight line of slope $-k_1$ and they would intercept at $\ln(q_e)$. A good agreement is shown for M-NS in the parameter values of $q_e$ and the rate constant in Table 1.

**Table 1.** Parameters for first and second-order kinetics.

|  | 1st Order Kinetics | | |
|---|---|---|---|
|  | **Intercept** | **Slope** | $R^2$ |
| Magnetic nano silica (M-NS) | 1.98028 | −0.8363 | 0.9103 |
|  | **2nd Order Kinetics** | | |
|  | **Intercept** | **Slope** | $R^2$ |
| Magnetic nano silica (M-NS) | 0.15816 | −0.08956 | 0.9116 |

The data is obtained from the slope and the intercepts of the curves. The first-order adsorption kinetics is given as:

$$\left(\frac{dq}{dt}\right) = k_2(q_e - q)^2,\tag{4}$$

where $k_2$ represents the rate constant given as the adsorbent (g) per contaminant (mg), per min. Integrating the above Equation (4) with the limit q = 0 at time t = 0, we have:

$$\left(\frac{1}{q_e - q}\right) - \left(\frac{1}{q_e}\right) = -k_2 t,\tag{5}$$

If we rearrange the equation, it becomes:

$$\left(\frac{t}{q}\right) = \left(\frac{t}{q_e}\right) + \left(\frac{1}{k_2 q_e^2}\right),\tag{6}$$

In this case, a plot of (t/q) vs. time would result in a straight line plot having a slope of $(1/q_e)$ and the y-intercept of $(k_2 q^2 e)^{-1}$. In this case, the fit is not as good as the one of the linear kinetics. The parameter values of the slope and intercept along with the goodness of fit, $R^2$, are also given in Table 2. With an $R^2$ of 0.9103, the first order can also be used to explain the process of the sorption. The first order kinetic would suggest that the sorption increases in relation to time and the amount of sorbent used, as seen in the sorption; this is true until the sorption equilibrium is reached. However, the second order kinetics suggest that the sorption process is related to the number of contaminants. It can also be used to understand the reason why sorption decreased slightly with the addition of adsorbent.

**Table 2.** Parameters for Langmuir and Freundlich isotherms.

|  | Langmuir | | |
|---|---|---|---|
|  | $R^2$ | **Qm** | **b** |
| Magnetic nano silica | 0.9964 | 0.342 | −2.8 |
|  | **Freundlich** | | |
|  | $R^2$ | *n* | $K_f$ |
| Magnetic nano silica | 0.9568 | −12.02 | 2.135 |

### 3.4. Adsorption Isotherm

The sorption isotherm relates the equilibrium concentration of the contaminant on the surface of the sorbent (qe) to the concentration of the contaminant in the contact liquid (Ce). In this study, the Freundlich and Langmuir isotherm was considered. The Freundlich isotherm can be represented as:

$$Q_e = KCe^{1/n},$$ (7)

In this case, the isotherm parameters are referred to as K and $n$, and they can be conveniently determined by rearranging the above equation as:

$$\ln(Q_e) = \frac{1}{n}\ln(Ce) + \ln(K),$$ (8)

The Langmuir isotherm is generally given as:

$$q_e = \frac{Q_m bCe}{b1 + bCed},$$ (9)

where the terms have their usual notations. The isotherm fit for both M-NS can be studied in the parameter values, as given in Table 2. As shown in Table 2, the Langmuir isotherm gave the best fit for the process of the adsorption with an $R^2$ of 0.9964. The Langmuir isotherm suggests that there is adsorption onto the surface of the magnetic nano silica and there is electrostatic interaction between the adsorbent and the adsorbate. However, with an $R^2$ of 0.9568 the Freundlich isotherm fit suggests the presence of multilayer adsorption onto the surface of the nanoparticle and the presence of weak force, such as van der Waal force [53].

### 3.5. Mechanism of Sorption

To fully understand the mechanism of the sorption, it is important to fundamentally understand the interactions that occur between the nanoparticle and the petroleum contaminants [54]. The negative surface charge associated with the magnetic silicon nanoparticles is important in the interaction between the adsorbent and adsorbate. Sorption requires a high interactive connection between sorbent and sorbate, usually brought about by electrostatic fields. The presence of negative sites in the magnetic nano silica suggests that there are electrostatic interactions. The sorption kinetics, however, suggest both physio and chemisorption mechanisms. With respect to the electrostatic interactions that exist between the synthesized magnetic nano silica and the petroleum-based contaminants, it appears that only surface adsorption occurs since the adsorbent is non-porous [55]. Furthermore, an electric field towards the surface of the silicon end of the nanoparticles creates a positive–negative attraction of molecules, as such attractions are orchestrated by the electric field [56].

### 3.6. Reusability Assessment

For a material to be classified as a sustainable material, it must meet some criteria of reusability. Reusable sorbents are more economical and are better to use since they can be used continuously [57]. The results in Figure 9 show the rate of efficiency of magnetic nano silica after five uses. The adsorbent remained uniformly potent at the second use, but at the third use the percentage of sorption dropped to 83%, and dropped to 80% percent for the fourth and fifth use. In most other studies conducted on the uptake of petroleum contaminants, the adsorbent was seldom reusable after the first use [58]. Thus, we can state that the magnetic nano silica is reusable, as only a 5% decrease was experienced in five circles of reuse for our studies.

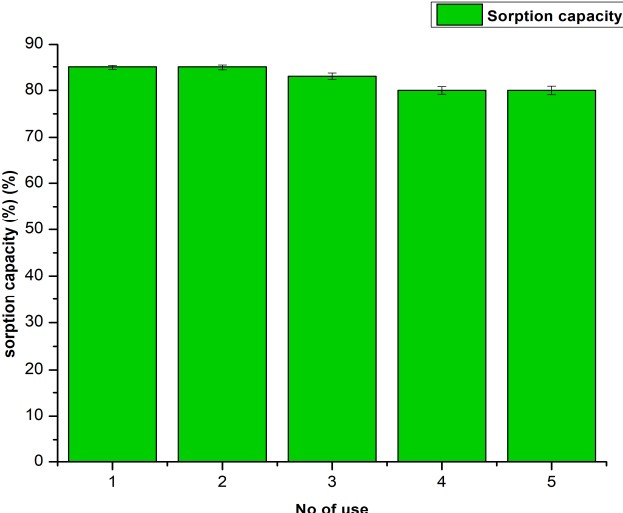

**Figure 9.** Reusability in studies of M-NS particles for the adsorption capacity of petroleum removal.

## 4. Conclusions

Magnetic nano silica particles were synthesized from barley husk waste through acid reflux and heat treatment processes. Characterization results revealed that the M-NS particles were 162 nm in size and had good stability. The study also showed that the particles were slightly amorphous and had non-porous structures. The synthesized magnetic nano silica was used to remove petroleum and its byproducts from water. To understand the amount of petrol contaminants that were removed, we conducted studies on the effects of pH, contact time, reusability, and weight of the sorbent. The kinetics and isotherm of the sorption reaction were also studied to understand the mechanism of the sorption. At neutral pH, the sorption was significantly effective, and the percentage of contaminant removed was seen to slightly depend on the weight of the sorbent and time until the sorption equilibrium was reached. Although the magnetic nano silica was synthesized through green synthesis, the removal efficiency of 85% was seen to be very high as compared to other adsorbents used in other studies. The magnetic property of the synthesized M-NS allowed for easy retrieval of the adsorbent using a sizable magnet. This provides an attractive alternative for the use of chemically produced adsorbents which may seem otherwise expensive. The special qualities of the adsorbent used in this study include a magnetic property that allows for easy retrieval of the adsorbent after use, and its sustainability as a result of green synthesis. To qualify as a sustainable material, it is important to understand the environmental sustainability of the synthesized M-NS. The environmental toxicity of the synthesized M-NS is of importance owing to the fact that the nano particles are synthesized from agricultural waste materials. Although it is beyond the scope of this study, the nanomaterials synthesized for this investigation present little to no eco-toxicity, due to their biocompatiblity and also due to their reduced dimensions, with remarkable chemical and biological activities and no threats to humans and the environment. We feel that this method, upon commercialization, has a great potential for environmental sustainability.

**Author Contributions:** Conceptualization, D.K., E.A. and A.V.; methodology, E.A.; software, E.A.; validation, D.K. and E.A.; formal analysis, E.A.; investigation, E.A.; resources, D.K.; data curation, E.A.; writing—original draft preparation, D.K., A.V. and E.A.; writing—review and editing, D.K., A.V. and E.A.; visualization, D.K., A.V. and E.A.; supervision, D.K.; project administration, D.K. All authors have read and agreed to the published version of the manuscript.

**Funding:** This research received no external funding.

**Conflicts of Interest:** The authors declare no conflict of interest.

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
