# Peer review of "Novel Magnetic Nano Silica Synthesis Using Barley Husk Waste for Removing Petroleum from Polluted Water for Environmental Sustainability"

_sustainability, doi:10.3390/su122410646_

Round 1
Reviewer 1 Report
The paper entitled “Novel Magnetic Nano Silica Synthesis Using Barley Husk Waste for Removing Petroleum from Polluted Water for Environmental Sustainability” by Akhayere et. al reports the synthesis of magnetic silica nanoparticles and their use in petroleum removal from polluted water.
The paper discussed the using of magnetic nanoparticles for removal of a petroleum product from water, this is an emerge need, but it has been reported in several papers, however the use of barley husk in synthesis of the silica particle gives the paper an advantage over previous papers.
The manuscript has been Witten in a technical report style rather than a scientific article in a peered review journal. Therefore, there are several points that need to be addressed before being suitable for publication in Sustainability.
- Synthesis of silica nanoparticles as reported is not clear and needs more discussion of the science behind converting the barely husk to a silica particles. Synthesis scheme doesn’t give the clear process starting from the synthesis of the nano particles.
- The role of barely leaves extract is not clear in synthesis of the magnetic NPs. Stirring Fe3O4 for long time could give the particle even with the absence of the barely leaves.
- FTIR spectra are not convincing, especially that the peak around 1200 (for Si-O) could be for both silica nanoparticles or for the silica-magnetic nanoparticles. And the bond Fe-O-Si is not clear, is it between the silica and magnetic FeNPs ?or the iron here is dropped as an iron ions not nanoparticles (just using the iron salt).
- All figures need to be in higher quality with a white or no background. Arrows and the text in the figures has a low quality resolution. Figures need to be done in a professional software (excel….), scheme using ChemDraw or other software.
- The photographs are not in a high resolution, and not consistent.
- SEM images are unclear and their scale bar is not clear as well.
- DLS results has been presented as an image rather that data, it would be better to be exported and a plot to be produced using a data analysis software, data on the figure can be discussed in the text. The standard deviation is large as seen from the figure which give an idea that the experimental process is not controlled
- The manuscript need major revision for English including, sentence structures and formatting. Figures in text sometimes been referred as figure, fig., and Figures …. Need to be consistent. Many sentences are hard to read and understand (e.g. line 108:”… changed to a very dark brown color, indicating, and the formation of magnetic…..)
Therefore, the manuscript as such is not acceptable for publication.
Author Response
Dear Reviewer, #1:
The authors wish to thank you for your valuable review and comments. Comments, such as these provide additional set of eyes to look over the gaps in the manuscript by the authors. We have taken all of your comments into consideration, as we are convinced that it is geared towards providing better understanding of the research for the readers. We hope that you will find our response acceptable. We remain deeply indebted to your help and at your disposal, should additional edits become necessary.
Reviewer 1
Synthesis of silica nanoparticles as reported is not clear and needs more discussion of the - science behind converting the barely husk to a silica particle. Synthesis scheme doesn’t give the clear process starting from the synthesis of the nano particles.
Answer: We thank the reviewer for this review, more details have been added to the synthesis in the manuscript. In the manuscript this part corrected as Synthesis of nanoparticles:
The barley husk was cut into smaller pieces and dried for seven days, then washed thoroughly using distilled water, this was done to remove sand, adhering soil and visible particles from the straws. The resulting sample was dried in the oven for 24hrs at 100 °C. The dried barley husk is then crushed to powdered form using a miller. 50 g of barley husk powder is then refluxed in 250 ml of 2M HCl for 6 hrs. After the acid refluxing the sample is filtered and washed in distilled water then heated at 700 °C for 5 hrs., the synthesis procedure for nano-silica is same as described in [30]. The resulting nano silica powder is used for the preparation of nano silica solution.
To synthesize Magnetic nano silica (M-NS), it was important to first prepare the magnetic solution and the nano silica solution. Iron (III) chloride (FeCl3) was used as the precursor for the synthesis of the magnetic nanoparticles. 20 mL of the barley leaf extract was added to 200 mL of 1 mM FeCl3 solution at ordinary temperature. The resultant mixture was stirred using a homogenization stirrer at 10,000 rpm for 60 min [31, 32]. During this process, the color of the mixture changed to a very dark brown as seen in Figure 2b, indicating, the formation of magnetic nanoparticles. The mixture is later centrifuged three times for 10 min each and washed with alcohol twice. Within this process the color of the mixture gets intensely darker as can be seen in Figure 2b and c.
Magnetic nano silica (M-NS) particles were synthesized with a mixture of the magnetic solution and nano-silica solution. The nano silica solution was prepared by dissolving 4g of barley synthesized nano-silica in 10ml distilled water and stained for 20min. Magnetic nano silica was prepared by slowly dropping the magnetic solution into the nano-silica solution with continuous mixing for 15min. After 15 min, the solution was centrifuged (Figure 2a) and dried in the oven at 80 °C for 3hrs.
The role of barely leaves extract is not clear in synthesis of the magnetic NPs. Stirring Fe3O4 for long time could give the particle even with the absence of the barely leaves.
Answer: we thank the reviewer for this review, for the purpose of our synthesis barley leaf extract. contains 4% Fe3O4 thus instead of using distilled water as a dilution fluid, adding barley leaf extract to Fe3O4 played the role of stabilizing and capping agent during the synthesis of the magnetic nanoparticles and also ensures more sustainability in the synthesis procedure.
FTIR spectra are not convincing, especially that the peak around 1200 (for Si-O) could be for both silica nanoparticles and for the silica-magnetic nanoparticles. And the bond Fe-O-Si is not clear, is it between the silica and magnetic FeNPs? or the iron here is dropped as an iron ion not nanoparticles (just using the iron salt).
Answer: we thank the reviewer for this question, the FTIR image has been redrawn in the manuscript for better understanding, however the Fe-O-Si exist between the iron nano particle and the nano silica in the formation of magnetic nano silica.
All figures need to be in higher quality with a white or no background. Arrows and the text in the figures have a low-quality resolution. Figures need to be done in a professional software (excel….), scheme using ChemDraw or other software.
Answer: we thank the reviewer for this review, the figures have been redrawn and more details have been highlighted in all the redrawn figures, we hope the new additions provides better understanding for readers.
The photographs are not in a high resolution, and not consistent. SEM images are unclear and their scale bar is not clear as well., DLS results has been presented as an image rather that data, it would be better to be exported and a plot to be produced using a data analysis software, data on the figure can be discussed in the text.
Answer: we thank the reviewer for this review, the high-resolution figures for SEM have been provided in the manuscript, also the size distribution data from the DLS has been professionally drawn while the other information has been discussed in the text. More details have been highlighted in all the redrawn figures, to ensure consistency, we hope the new additions provides better understanding for readers.
The manuscript needs major revision for English including, sentence structures and formatting. Figures in text sometimes been referred as figure, fig., and Figures …. Need to be consistent. Many sentences are hard to read and understand (e.g. line 108:” … changed to a very dark brown color, indicating, and the formation of magnetic…...)
Answer: we thank the reviewer for this suggestion, the manuscript has been proofread to provide more consistency in the English, and we hope the reviewer finds the new additions sufficient.
Reviewer 2 Report
Please see the attachment

Author Response
Dear Reviewer, #2:
The authors wish to thank you for your valuable review and comments. Comments, such as these provide additional set of eyes to look over the gaps in the manuscript by the authors. We have taken all of your comments into consideration, as we are convinced that it is geared towards providing better understanding of the research for the readers. We hope that you will find our response acceptable. We remain deeply indebted to your help and at your disposal, should additional edits become necessary.
Please see the attachment.

Reviewer 3 Report
In order to show the efficiency of the synthesized MNS, it is recommended to compare its performance with other sorbents
Ln 105-111 & Ln 114-120: there are two different methods described in these parts of themanuscript and it is quite confusing which nanoparticles were used in the experiments. Also, there are different terms used for nano particles,for example, magnetic nano particle, magnetic silica nakoparticle, magnetic nanosilica,that seeme to indicate the same thing. Please eitherdefine each term more clearly if they meant to indicate different things or be consistent with the terms used in the manuscript.
There are no explanation about Figure 3A and 3B. Also, no explanation for Figure 6.
No A, B labelling in Figure 4
Ln 231-237: the fastest sorption was observed between 2 and 4 minutes according to Figure 8A. But the mamuscript says different things. Please revise the manuscript.
Ln256-257: I don't see any significant reduction in sorption rates in Figure 8C. Please revise this part of the maniscript.
Figure 8B and 8C: it could mislead the readers if the dots are connected by curved lines. Also please show error bars im the figures and indicate how many replicates were used for each experiment.
There is no discussion on thr sorption kinetics and isotherms. The results need to be discussed!
The equation numbers need to be revised.
English needs to be revised throughout the manuscript. There are many places that are hard to understand.
Please remove figure title in the figure 9.
Ln 339-340: in order to say that the synthesized sorbent is cost-effective, there needs to be some data on cost or cost analysis results that can support the statement.
Ln 345-347: the last statment is not appropriate without any cost analysis or any scale-up studies. Please revise the conclusions.
Author Response
Dear Reviewer, #3:
The authors wish to thank you for your valuable review and comments. Comments, such as these provide additional set of eyes to look over the gaps in the manuscript by the authors. We have taken all of your comments into consideration, as we are convinced that it is geared towards providing better understanding of the research for the readers. We hope that you will find our response acceptable. We remain deeply indebted to your help and at your disposal, should additional edits become necessary.
Reviewer 3
Ln 105-111 & Ln 114-120: there are two different methods described in these parts of the manuscript and it is quite confusing which nanoparticles were used in the experiments. Also, there are different terms used for nano particles, for example, magnetic nano particle, magnetic silica nanoparticle, magnetic nano silica, that seem e to indicate the same thing. Please either define each term more clearly if they meant to indicate different things or be consistent with the terms used in the manuscript.
Answer: we thank the reviewer for this review, the synthesis methods described are both for nano silica, and magnetic nano silica respectively, and the term magnetic nano silica particle (M-NS) has been used evenly to represent the synthesized adsorbent. However more clarity has been provided in the manuscript according to the reviewer’s recommendation and to better explain the respective images.
There is no explanation about Figure 3A and 3B. Also, no explanation for Figure 6. No A, B labelling in Figure 4
Answer: we thank the reviewer for this review, explanation of Figure 3and b are given in the sorption experiment page as follows “The sorption experiments were performed in batch as shown in Figure 3A and B, to ascertain how effective the synthesized magnetic nano silica was for the removal of petrol from contaminated water. 0.6g of Magnetic nano-silica (M-NS) was added to a beaker containing 500ml of petrol contaminated water. The concentration of the petrol contaminant was 40mg/L. Figure 3a shows the contaminated water before sorption while Figure 3b shows the water after sorption. The effect of contact time between the nanoparticles and the contaminant was first observed, the experiment contact was allowed for 2, 4, 6, 8, 10, 12, 14, 16, 18 20, and 22 minutes respectively, maximum sorption was observed at 12 minutes, which was the time at which equilibrium was reached. The optimization of pH was carried out by adjusting the pH of the system at different pH values 2–10, the samples were then analyzed to obtain the most suitable pH for the experiment.” Also Figure number has been provided for Figure 6, and adequate labelling for Figure 4.
Ln 231-237: the fastest sorption was observed between 2 and 4 minutes according to Figure 8A. But the manuscript says different things. Please revise the manuscript.
Answer: we thank the reviewer for this review, this statement has been rephrased to “The highest uptake of 85% was achieved at 12min and the rate was constantly maintained until the experiment was concluded at 22minutes. This shows that the equilibrium time for sorption of petrol contaminants using magnetic nano silica is 12minutes. In experiments conducted by [47], similar sorption equilibrium time was attained. The progression of the sorption experiment in relation to time shows that sorption increased as time increased until the equilibrium time is reached.”
Ln256-257: I don't see any significant reduction in sorption rates in Figure 8C. Please revise this part of the manuscript.
Answer: we thank the reviewer for this review, the line has been rephrased as follows: “The weight of sorbent with respect to the amount of petrol adsorbed was recorded and is as shown in Figure 8c. It is observed that the highest uptake result was gotten at 0.6g sorbent weight in correspondence with studies by [50,51] which suggests that sorption increases with respect to the weight of sorbent, we can also see that sorption increase with respect to the amount of sorbent available. Sorption rate also reduced to 83% at a sorbent dosage of 0.8g, as against 85% earlier experienced.” This information is as described in the image.
Figure 8B and 8C: it could mislead the readers if the dots are connected by curved lines. Also please show error bars in the figures and indicate how many replicates were used for each experiment.
Answer: we thank the reviewer for this suggestion, we see the need to connect the Figure 8 to better explain the progression of the sorption, according to studies by Syed S 2015, Syed and Alhazzaa 2011, Akhayere et al 2019 and 2020.
There is no discussion on the sorption kinetics and isotherms.
Answer: we thank the reviewer for this review, “With an R2 of 0.9103 the first order can also be used to explain the process of the sorption. The first order kinetic would suggest that the sorption increases in relations to time and the amount of sorbent used, as seen in the sorption, this is true until the sorption equilibrium is reached. However, the second order kinetics which suggest that the sorption process will be related to the number of contaminants can also be used to understand the reason why sorption slightly decreased with the addition of more adsorbent.”
“As shown in Table 2 the Langmuir isotherm gave the best fit for the process of the adsorption with an R2 of 0.9964. The Langmuir isotherm suggest that there is adsorption onto the surface of the magnetic nano silica and there is electrostatic interaction between the adsorbent and the adsorbate. However, with an R2 of 0.9568 the Freundlich isotherm fit suggests the presence of multilayer adsorption onto the surface of the nanoparticle and the presence of weak force such as van der Waal forces [52].”
English needs to be revised throughout the manuscript. There are many places that are hard to understand.
Answer: we thank the reviewer for this suggestion, the English errors in the manuscript have been revised and corrected, and we hope it is more explainable for the readers.
Please remove figure title in the figure 9.
Answer: we thank the reviewer for this suggestion, the title for Figure 9 has been corrected
Ln 339-340: in order to say that the synthesized sorbent is cost-effective, there needs to be some data on cost or cost analysis results that can support the statement. Ln 345-347: the last statement is not appropriate without any cost analysis or any scale-up studies. Please revise the conclusions.
Answer: we thank the reviewer for this review, however cost effective was related to how the materials for the synthesis was obtained, since the needed materials were not purchase rather produced trough a sustainable means of waste conversion, we thought it wise to state the cost effectiveness.
Reviewer 4 Report
The paper presents sustainable transformation of barley husk waste into magnetic silica nanoparticles, and its application for the removal of petroleum from water.
The paper can be published after major corrections and my recommendations are:
- The overall design strategy (choice of the specific materials for the chosen application) should be better explained in the final part of the introduction, based on the reported references.
- The characterization chapter must be improved. The authors must offer details regarding the operation conditions for each equipment.
- It should be underline how the materials were prepared
- The XRD pattern of the sample must be better explained
- The characterization is poorly discussed and some sentences are not (or are poorly) supported by experimental data.
- It is not clear how was measured the concentration of petroleum in water (before and after the sorption tests). Please indicate the method, equipment, etc.
- For almost the figures, they have low resolutions. Please modified them
- For chapter 3.3 the use of equation can not replace discussion. The explanation of equations is redundant and must be removed from the text. The authors must offer scientific explanations for their experiments.
- The same problem as the above is for chapter 3.4. Please offer scientific explanations for your experiment.
- The mechanism for pollutant adsorption must be explained
- The conclusion chapter must be replaced with some more scientifically conclusions.
- Some mistakes in English spelling and grammar should be corrected
Author Response
Reviewer 4
Dear Reviewer, #4:
The authors wish to thank you for your valuable review and comments. Comments, such as these provide additional set of eyes to look over the gaps in the manuscript by the authors. We have taken all of your comments into consideration, as we are convinced that it is geared towards providing better understanding of the research for the readers. We hope that you will find our response acceptable. We remain deeply indebted to your help and at your disposal, should additional edits become necessary.
The overall design strategy (choice of the specific materials for the chosen application) should be better explained in the final part of the introduction, based on the reported references.
Answer: we thank the reviewer for this suggestion, “Several researches have explored the use of chemically produced nano silica as adsorbent for the removal of oily contaminants in water, but in terms of sustainability the production of such kind of nano silica do not meet the green chemical synthesis. Nano silica synthesized in this study was achieved through green synthesis, the idea of using a waste material for its synthesis was geared towards achieving a sustainable environment. The magnetization of the nano silica was to allow easy collection of the adsorbent after adsorption, by the simple use of a magnet. To suit the ideology, the research was designed such that nano silica was first synthesized, then iron nano particles was synthesized, and lastly nano silica and iron nano particles was used for the synthesis of magnetic nano silica. The magnetic nano silica was then used for the removal of petrol contaminants from water”. These statements have been added to the introduction according to the reviewer’s suggestion.
The characterization chapter must be improved. The authors must offer details regarding the operation conditions for each equipment. It should be underlining how the materials were prepared
Answer: we thank the reviewer for this suggestion, the characterization chapters has been duly improved with more details according to the reviewer’s suggestion.
The XRD pattern of the sample must be better explained
Answer: we thank the reviewer for this suggestion, the XRD image has been made clearer and the following explanatory details have been provided: “The X-ray pattern of Magnetic nano-silica (M-NS) particles is shown in Figure. 4A. The diffraction peaks of synthesized Fe3O4-NPs shown in Fig. 4A-c were detected at 2θ = 30.6°, 35.9°, 43.5°, 54.0°, 57.4°, 63.00°, and 74.5°, these were allocated to the crystal planes of (200), (311), (400), (422), (511), (440), and (533), respectively. The diffraction peaks as analyzed, corresponded with the standard magnetite XRD patterns according to Joint Committee on Powder Diffraction Standards (JCPDS) file no: 00-003-0863, which states the cubic structure crystallographic system. We can calculate the crystallite size of the synthesized Fe3O4-NPs with the help of the Debye-Scherrer equation which reveals a relationship between X-ray diffraction peak broadening and crystallite size [34]. The estimated crystallite size of synthesized Fe3O4-NPs was 10.12 nm, this is calculated from the full-width at half maximum of the crystal planes of (311) [35]. According to the XRD pattern, the synthesized Fe3O4-NPs were seen to be purely crystalline lean with little or no notable impurity peaks.
Subsequently XRD peak of nano silica was given at 2θ = 20-30°. The result as shown in Fig. 4A-b revealed that nano silica particles were amorphous, however the result also revealed that there were very few impurities as a result of other chemical elements present in the barley husk waste, which was the raw material for synthesis of nano silica. The highest peak of the nano-silica particle was observed at 2θ=24.8º. According to the XRD results it is valid to say that the method of synthesis of nano-silica from barley significantly resulted in particles with low impurities [16, 30]. Lastly it can be seen from the XRD result that there were no substantial changes observed in case of the peaks for magnetic nano silica as shown in Fig. 4A-a. Although a new peak was identified at 2θ = 20-30° which indicates the presence of amorphous silica. Similar results were observed in XRD patterns obtained by Munasir and Terraningtyas, 2019 [36]. “
The characterization is poorly discussed, and some sentences are not (or are poorly) supported by experimental data.
Answer: we thank the reviewer for this review, more discussions have been provided for the characterization of the nano particles and the figures have been made clearer to provide better understanding for the readers.
It is not clear how was measured the concentration of petroleum in water (before and after the sorption tests). Please indicate the method, equipment, etc.
Answer: we thank the reviewer for this review, “The amount of adsorbent used for sorption is a very important parameter, to get the required amount of sorbent, for effective sorption of petrol, different amounts were used on the contaminated water at a neutral pH, [33] with the adequate contact time, and the samples were then withdrawn for analysis in the OCMA 310 Oil Content Analyzer. The oil content analyzer first extracts the oil using a solvent (Polychlorotrifluoroethylene) and then analyze it by infrared spectrophotometry. The result is given mg/l oil on a digital panel meter.”
For chapter 3.3 the use of equation cannot replace discussion. The explanation of equations is redundant and must be removed from the text. The authors must offer scientific explanations for their experiments.
Answer: we thank the reviewer for this review, we feel the need to explain the equation for better understanding of readers however more discussions have been provided for the kinetics and isotherms in the manuscript as follows: we thank the reviewer for this review, the sorption kinetics and isotherms are discussed as follows “With an R2 of 0.9103 the first order can also be used to explain the process of the sorption. The first order kinetic would suggest that the sorption increases in relations to time and the amount of sorbent used, as seen in the sorption, this is true until the sorption equilibrium is reached. However, the second order kinetics which suggest that the sorption process will be related to the number of contaminants can also be used to understand the reason why sorption slightly decreased with the addition of more adsorbent.”
“As shown in Table 2 the Langmuir isotherm gave the best fit for the process of the adsorption with an R2 of 0.9964. The Langmuir isotherm suggest that there is adsorption onto the surface of the magnetic nano silica and there is electrostatic interaction between the adsorbent and the adsorbate. However, with an R2 of 0.9568 the Freundlich isotherm fit suggests the presence of multilayer adsorption onto the surface of the nanoparticle and the presence of weak force such as van der Waal forces [52]”.
The same problem as the above is for chapter 3.4. Please offer scientific explanations for your experiment.
Answer: we thank the reviewer for this review, correction have been made as above.
The mechanism for pollutant adsorption must be explained
Answer: We thank the reviewer for this review, the negative surface charge area in the magnetic silicon nanoparticles is important in the interaction between the adsorbent and adsorbate. Sorption requires a high interactive connection between sorbent and the sorbate, usually brought about by forces and electrostatic fields. The presence of negative sites in the magnetic silica nanoparticles suggests that there are a physio and chemisorption mechanism, concerning the electrostatic interactions that exist between the synthesized magnetic silica nanoparticles and the petrol contaminant in the aqueous environment.
The conclusion chapter must be replaced with some more scientifically conclusions.
Answer: we thank the reviewer for this review, more scientific details have been provided on the conclusion as follows “Magnetic Nano-silica was synthesized from barley husk waste through acid treatment and heat treatment. Characterization results revealed that the nano particles were 162nm in sizes and had good stability, it also showed that the particles were slightly amorphous and had no pores in their structure. The synthesized magnetic Nano silica has been used to remove petrol from water. To understand the amount of petrol contaminants that were removed the effects of pH contact time, reusability, and weight of sorbent were studied. Kinetics and isotherm of the sorption reaction was also studied to understand the mechanism of the sorption. At neutral pH, the sorption was significantly effective, and the percentage of contaminant removed was seen to slightly depend on the weight of the sorbent and time until the sorption equilibrium is reached. Although the magnetic nano particle was synthesized through green synthesis, the removal efficiency of 85% was seen to be very high as compared to other adsorbent used in other studies. The magnetic property of the nano-silica allowed for easy retrieval of the adsorbent using a sizable magnet. This provides an alternative for the use of chemically produced adsorbents which are sometimes expensive the special qualities of the adsorbent include magnetic property that allow for easy retrieval of the adsorbent after use. To qualify for a sustainable material, it is important to understand environmental sustainability of the synthesized M-NS. The material has been synthesized from agricultural waste thus promotes the zero-waste environmental sustainability goal. Nanomaterials generally have little or no eco-toxic influences, with very remarkable chemical and biological activities and pose little or no threats to human and environment”.
Some mistakes in English spelling and grammar should be corrected
Answer: we thank the reviewer for this review, English corrections have been made in the manuscript.
Reviewer 5 Report
In the submitted manuscript the authors considered the preparation of magnetic silica nanoparticles by using barley waste material. In general, the experimental characterization was properly conducted and all the results sufficiently commented. However, this manuscript has limited interest to the readers of Sustainability, being the sustainable aspect limited to the use of a waste that was acid treated to recover silica. Few points here listed has limited the bouncy of the paper, that should submitted to another journal:
- The adsorption kinetics evaluation is, in my opinion, out of the scope of the journal
- it is not clear to the reader how the extract was used for the preparation of the magnetic nanoparticles
- no information have been included in the introduction on the use of natural extract to synthetize these (or similar) particles;
- the sustainability is limited if we think that at the third reuse, the efficiency strongly decreased
- how the magnetic nanoparticle work when in contact with the petroleum contaminated solution? The absorption was studied as a function of contact time, pH and sorbent dosage, but how the authors can justify the efficiency of the absorber from its chemical structure? It is not clear at all how the removal was successful. If the removal is only due to the magnetism, please justify the reason to use silica based magnetic nanoparticles instead of neat iron oxide nanoparticles;
- Could the authors prove the electrostatic interactions that should be responsible of the absorption?
Author Response
Dear Reviewer, #5:
The authors wish to thank you for your valuable review and comments. Comments, such as these provide additional set of eyes to look over the gaps in the manuscript by the authors. We have taken all of your comments into consideration, as we are convinced that it is geared towards providing better understanding of the research for the readers. We hope that you will find our response acceptable. We remain deeply indebted to your help and at your disposal, should additional edits become necessary.
Reviewer 5
it is not clear to the reader how the extract was used for the preparation of the magnetic nanoparticles
Answer: we thank the reviewer for this review, the manuscript provides details on the synthesis methodology, however, to suit the reviewer’s need we have explained in more details how the synthesis was carried out, as follows. “The barley husk was cut into smaller pieces and dried for seven days, then washed thoroughly using distilled water, this was done to remove sand, adhering soil and visible particles from the straws. The resulting sample was dried in the oven for 24hrs at 100 °C. The dried barley husk is then crushed to powdered form using a miller. 50 g of barley husk powder is then refluxed in 250 ml of 2M HCl for 6 hrs. After the acid refluxing the sample is filtered and washed in distilled water then heated at 700 °C for 5 hrs., the synthesis procedure for nano-silica is same as described in [30]. The resulting nano silica powder is used for the preparation of nano silica solution.
To synthesize Magnetic nano silica (M-NS), it was important to first prepare the magnetic solution and the nano silica solution. Iron (III) chloride (FeCl3) was used as the precursor for the synthesis of the magnetic nanoparticles. 20 mL of the barley leaf extract was added to 200 mL of 1 mM FeCl3 solution at ordinary temperature. The resultant mixture was stirred using a homogenization stirrer at 10,000 rpm for 60 min [31, 32]. During this process, the color of the mixture changed to a very dark brown as seen in Figure 2b, indicating, the formation of magnetic nanoparticles. The mixture is later centrifuged three times for 10 min each and washed with alcohol twice. Within this process the color of the mixture gets intensely darker as can be seen in Figure 2b and c.
Magnetic nano silica (M-NS) particles were synthesized with a mixture of the magnetic solution and nano-silica solution. The nano silica solution was prepared by dissolving 4g of barley synthesized nano-silica in 10ml distilled water and stained for 20min. Magnetic nano silica was prepared by slowly dropping the magnetic solution into the nano-silica solution with continuous mixing for 15min. After 15 min, the solution was centrifuged (Figure 2a) and dried in the oven at 80 °C for 3hrs.”
no information has been included in the introduction on the use of natural extract to synthetize these (or similar) particles;
Answer: we thank the reviewer for this review, more details have been provided on the introduction according to the reviewer’s recommendation.
the sustainability is limited if we think that at the third reuse, the efficiency strongly decreased
Answer: we thank the reviewer for this review, the adsorbent remained uniformly potent at the second use, but at the third use the percentage sorption dropped to 83%, and 80% percent for the fourth and fifth use. in most studies conducted on uptake of petroleum contaminants, adsorbent is seldom reusable after first use we can state that the magnetic nano particle is reusable thus sustainable because only 5% declination is experienced in five circles of reuse, also most adsorbent is hardly reusable after contact with petroleum hydrocarbon, and lastly the particles were synthesized via green synthesis.
how the magnetic nanoparticle work when in contact with the petroleum contaminated solution? The absorption was studied as a function of contact time, pH and sorbent dosage, but how the authors can justify the efficiency of the absorber from its chemical structure? It is not clear at all how the removal was successful. If the removal is only due to the magnetism, please justify the reason to use silica based magnetic nanoparticles instead of neat iron oxide nanoparticles;
Answer: we thank the reviewer for this review, the reaction was an adsorption reaction thus the mechanism was based on adsorption not absorption, this was made successful by the negative surface charge area in the magnetic silicon nanoparticles which aids the interaction between the adsorbent and adsorbate. Adsorption requires a high interactive connection between sorbent and the sorbate, usually brought about by forces and electrostatic fields. The presence of negative sites in the magnetic silica nanoparticles suggests that there are a physio and chemisorption mechanism, concerning the electrostatic interactions that exist between the synthesized magnetic silica nanoparticles and the petrol contaminant in the aqueous environment. The magnetization of silica was to aid the collection of the adsorption after sorption has taken place however the contaminants are not adsorbed due to the magnets.
Could the authors prove the interactions that should be responsible of the absorption?
Answer: yes, we can. As shown in the manuscript, the contaminants were removed by an adsorption reaction. The adsorption would not take place without the existence of the electrostatic interactions as there are no pores formation when the nano particles were characterized. Also, the kinetics and isotherm results give better understanding of how the adsorption is taking place.
Round 2
Reviewer 1 Report
Although the manuscript has been improved, still doesn't meet the high level in term of science.
- the science behind using the barely in this experiment still not clear, why barely, why not other similar sources? what is the chemistry in barely that make it special?
- Figures are still not in a high quality to be published in a scientific journal, there is no consistency in the font format, size clarity,.....
- some added information, like that at the end of page 2, doesn't have any evidences and not supported by references and looks like as an opinion
The manuscript need to take care of the science behind synthesis and compare this science (chemistry)with conventional methods to support the sustainability claim
Author Response
The authors are most appreciative of the comments of the reviewer.
Reviewer 1
The science behind using the barely in this experiment still not clear, why barely, why not other similar sources? what is the chemistry in barely that make it special?
Answer: we thank the reviewer for this kind question. We have added explanation to support the rational and what the chemistry special.
Barley is a plant that is largely cultivated in Cyprus and across the Mediterranean areas. This would also mean that barley generates the largest number of agricultural wastes in these areas. Barley husk (BH) is an example of an agricultural waste product, generated in areas where barley is largely cultivated [4, 5]. It can also be described as a popular waste that is generated during the harvest and usage of barley plants [6]. In Mediterranean areas like North Cyprus, where agriculture is a very large part of the everyday life, there is a high rate of the generation of barley waste, due to the high rate of barley cultivation, this is also true for other agriculturally intensive countries. Most times the farmers have no idea what to do with these wastes so they are left unattended to thereby creating as environmental issue. [7]. Barley plant belongs to the grass family and it is the fourth-largest grain crop grown around the world. As a grass plant, there is close contact between the plant and the soil this ensures a high silicon presence in the plant [8]. In research conducted by Akhayere et al. and Azizi et al. (XRF results showed that silicon was the compound with the highest presence in barley, this factor, makes it a good raw material for the production of silicon-related substances [9, 10].
Figures are still not in a high quality to be published in a scientific journal, there is no consistency in the font format, size clarity,
Answer: we thank the reviewer for this kind observation. Please note that all of the figures were reworked are now in high resolution. We have included all figures in high resolution which can be adjusted in size during the production phase.
some added information, like that at the end of page 2, doesn't have any evidences and not supported by references and looks like as an opinion
Answer: we thank the reviewer for this kind observation, we have added some references to the statement described.
The manuscript needs to take care of the science behind synthesis and compare this science (chemistry)with conventional methods to support the sustainability claim.
Answer: Indeed. We have taken care of this step as well.
Reviewer 3 Report
The comments were mostly reflected in the revised version of the manuscript; however, it is still not clear how many duplicates the authors used in the experiments.
Author Response
The authors are most appreciative of the comments of the reviewer.
The comments were mostly reflected in the revised version of the manuscript; however, it is still not clear how many duplicates the authors used in the experiments
Answer we thank the reviewer for this kind review, the experiments were repeated in 6 duplicates for all the parameters to ensure the accuracy of the results. This information has also been provided in the manuscript.
Reviewer 4 Report
I consider that the manuscript can be published after minor English corrections.
Author Response
The authors are most appreciative of the comments of the reviewer.
I consider that the manuscript can be published after minor English corrections
Response: We thank the reviewer for this kind comments.
We have gone through 5 additional revisions to ensure that all of the sentences are carefully articulated. I hope that you will find this version satisfactory, if not perfect.
Reviewer 5 Report
The overall quality of the manuscript has been improved, all the suggested changes have been revised and proper modification of the paper has been taken into account. I would ask again to improve the quality of the figures, that need to be replaced with more defined images (as in the case of Figure 1). In Figure 4B please reorganize the label position for x - axis, deviation for sorption capability values in Figure 9 has to be provided.
Author Response
The authors are most appreciative of the comments of the reviewer.
The overall quality of the manuscript has been improved, all the suggested changes have been revised and proper modification of the paper has been taken into account. I would ask again to improve the quality of the figures, that need to be replaced with more defined images (as in the case of Figure 1). In Figure 4B please reorganize the label position for x - axis, deviation for sorption capability values in Figure 9 has to be provided.
Answer: we thank the reviewer for this kind suggestion, image quality have been improved, label position for x - axis, in figure 4b has been reorganized and the standard deviation for sorption capability has been included in figure 9.
The authors thank the reviewers for the kind suggestions, comments and reviews, we are very confident that it is aimed at making the publication interesting to the readers, we hope that the reviewer find the corrections sufficient and to your satisfaction.
Round 3
Reviewer 1 Report
The manuscript has been improved. Text in Figures still need to be clearer.
The manuscript can be published after providing a clear text in Figures